# Survival Bias, Non-Lineal Behavioral and Cortico-Limbic Neuropathological Signatures in 3xTg-AD Mice for Alzheimer’s Disease from Premorbid to Advanced Stages and Compared to Normal Aging

**DOI:** 10.3390/ijms241813796

**Published:** 2023-09-07

**Authors:** Aida Muntsant, Maria del Mar Castillo-Ruiz, Lydia Giménez-Llort

**Affiliations:** 1Department of Psychiatry and Forensic Medicine, School of Medicine, Universitat Autònoma de Barcelona, 08193 Barcelona, Spain; aida.muntsant@uab.cat; 2Institut de Neurociències, Universitat Autònoma de Barcelona, 08193 Barcelona, Spain; mariadelmar.castillo@uab.cat

**Keywords:** aging, Alzheimer’s disease, heterogeneity, cognitive deficits, neuropsychiatric-like symptoms, amygdala, ventral hippocampus, animal model, social isolation

## Abstract

Pre-clinical research in aging is hampered by the scarcity of studies modeling its heterogeneity and complexity forged by pathophysiological conditions throughout the life cycle and under the sex perspective. In the case of Alzheimer’s disease, the leading cause of dementia in older adults, we recently described in female wildtype and APP23 mice a survival bias and non-linear chronology of behavioral signatures from middle age to long life. Here, we present a comprehensive and multidimensional (physical, cognitive, and neuropsychiatric-like symptoms) screening and underlying neuropathological signatures in male and female 3xTg-AD mice at 2, 4, 6, 12, and 16 months of age and compared to their non-transgenic counterparts with gold-standard C57BL/6J background. Most variables studied detected age-related differences, whereas the genotype factor was specific to horizontal and vertical activities, thigmotaxis, coping with stress strategies, working memory, and frailty index. A sex effect was predominantly observed in classical emotional variables and physical status. Sixteen-month-old mice exhibited non-linear age- and genotype-dependent behavioral signatures, with higher heterogeneity in females, and worsened in naturalistically isolated males, suggesting distinct compensatory mechanisms and survival bias. The underlying temporal and spatial progression of Aβ and tau pathologies pointed to a relevant cortico-limbic substrate roadmap: premorbid intracellular Aβ immunoreactivity and pSer202/pThr205 tau phosphorylation in the amygdala and ventral hippocampus, and the entorhinal cortex and ventral hippocampus as the areas most affected by Aβ plaques. Therefore, depicting phenotypic signatures and neuropathological correlates can be critical to unveiling preventive/therapeutic research and intervention windows and studying adaptative behaviors and maladaptive responses relevant to psychopathology.

## 1. Introduction

Neurological disorders are recognized as the principal cause of disability-adjusted life years [1]. Alzheimer’s disease (AD) is a heterogenous and multi-factorial neurodegenerative disorder caused by the interaction of demographic–environmental and genetic factors [2]. AD represents the leading causes of dementia, with age-related changes playing a crucial role [3]. Several studies highlight the heterogenous and complex clinical patterns and temporal progression, especially in the late stages of the disease [4,5]. Nowadays, there is a high failure rate in developing new disease-modifying treatments, and therapies seem to have limited benefits regarding AD pathogenesis and progression. More precise diagnosis criteria, recognition of AD heterogeneity, and personalized intervention strategies should improve outcomes in basic research and clinical trials [6].

At the clinical level, besides cognitive impairments and memory loss, additional clinical manifestations related to neuropsychiatric symptoms (NPS), also referred to as behavioral and psychological symptoms of dementia (BPSD), are observed in approximately 90% of patients. These include agitation, aggression (verbal or physical), irritability, anxiety, depression or other disruptive behaviors, sleep disorders, and hallucinations [7,8]. Age, gender, and comorbidities may influence the severity of neuropsychiatric symptoms [9]. Their management is challenging and usually requires pharmacological treatments (i.e., antipsychotics) and non-pharmacological intervention [10]. Moreover, NPS are the symptoms most related to disease burden, quality of life, and caregiver burnout [11].

At the neuropathological level, AD is mainly defined by extracellular accumulation of amyloid-β (Aβ) plaques and hyperphosphorylated tau proteins forming intracellular neurofibrillary tangles (NFTs), accompanied by reactive gliosis and synaptic dysfunction [12,13,14].

Moreover, although the connections are not yet fully understood, there is growing experimental, clinical, and epidemiological evidence that brain and periphery crosstalk interaction may be related to AD development and progression. Underlying systemic disease processes are reflected in AD patients: systemic immunity disorders, cardiovascular disease, hepatic and renal dysfunction, metabolic disorders, blood abnormalities, respiratory and sleep disorders, microbiota disorders, and systemic inflammation [15,16]. AD pathology is characterized by cortical atrophy, particularly in the multimodal association cortices and the structures of the limbic lobe, and the progressive degeneration of neurons in various areas (such as the hippocampus, amygdala, entorhinal, and neocortex) [14,17,18]. Regarding neuropsychiatric symptoms, the pathological mechanisms are not already known. However, the emotional and anxiety behaviors presented in AD have been associated with metabolic and volumetric alterations in the amygdala [19,20]. On the other hand, several articles have described the heterogeneous function of the rodent hippocampus, with the dorsal hippocampus playing an important role in cognitive function and connected to the brain regions connected to spatial memory; the ventral hippocampus has been related to anxiety- and stress-involved areas, i.e., connections between the amygdala and the ventral hippocampus and prefrontal cortex have been documented with anxiety-like behaviors and fear response in animal models [21,22,23,24].

At the translational level, the shorter lifespan of most animal models affords a fleet-footed scenario for long-term monitoring. For this reason, animal model characterization can help to better understand disease mechanisms from morphological to behavioral levels and to evaluate the safety and efficacy of new therapeutic approaches. Experimental gerontologists highlight the relevance of using aged animals to mimic humans’ complex and multi-factorial aging processes [25,26]. Moreover, the heterogeneity presented in AD patients has also been demonstrated in animal models for the disease, especially at advanced stages [27,28,29]. Studying the disease and age interaction effects in phenotypic characteristics using AD models can also provide information on normal and pathological aging processes. Among the animal models of AD, we have proposed asymptomatic, prodromal, onset, advanced, and long-term survivors of widely used 3xTg-AD mice. Based on the familial AD mutations PS1/M146V and APPSwe, and harboring tauP301L human transgene, this model progressively develops temporal- and regional-specific neuropathological patterns observed in the human brain of AD patients [30,31,32]. Cognitive deficits [33,34] and a broad spectrum of NPS-like symptoms have also been described [35,36]. In the initial reports of the 3xTg-AD model, mice first develop intraneuronal Ab at 3–4 months of age, followed by plaque formation at 6 months of age in the cortex and hippocampus, with NFT becoming apparent at 12 months of age [30,31]. However, in recent years, several studies have indicated a drift in the phenotype of mice, with males being particularly affected. This fact suggests that the widespread use of these mice has resulted in a generation of sublines with different onset and progression of neuropathology, causing controversy and confusion. Moreover, 3xTg-AD mice exhibit some variability even between littermates [32,37].

Here, we described the age, AD genotype, and sex sensitivity of a behavioral test battery in 3xTg-AD mice of male and female mice at asymptomatic (2 months), prodromal (4 months), onset (6 months), advanced (12 months), and very old (16 months) ages and compared them to age-matched non-transgenic mice with normal aging. Moreover, we assessed the spatial and temporal progression of Aβ pathology and tau hyperphosphorylation and studied the sex-dependent differences.

## 2. Results

### 2.1. Strong Contribution of Age but Not Genotype and Sex Effects to Target Variables for AD-Related Phenotype

Table 1 summarizes the effects of the main factors: genotype (G), sex (S), and age (A). As shown in most behavioral variables, behavioral performances strongly depended on age. Genotypes differed in their horizontal and vertical activities, thigmotaxis, coping with stress strategies, working memory, and frailty index. Sex effect was consistently observed in urination, a classical emotional variable, and sexual dimorphism in body weight. Also, sex differences were found in the horizontal and vertical activities in the corner and the open-field tests, on risk assessment as in the latency to cross the intersection in the T-maze, and coping strategies as in the distance covered in the CUE, the first paradigm in the MWM, and the frailty index. Interaction effects were also observed, with special mention given to the marble test and mostly involving age × sex interaction effects.

In the corner test, the horizontal locomotor activity (Figure 1A), measured by the number of visited corners, was lower in 3xTg-AD mice than in NTg animals (G*, *p* = 0.020). However, an age effect pointed to 4- and 12-month-old animals being more active compared to the group of 16-month-old animals (A***, *p* = 0.001). These differences were mainly observed in males. Sex differences were also observed, as females presented more visited corners (S**, *p* = 0.003). In the case of vertical activity, no statistical differences were observed in either the total number of rearings (Figure 1B) or the latency to perform the first rearing (Figure 1C).

The ethogram describing the temporal sequence of behavioral events of animals in the open-field test (Figure 2A) showed that the actions exhibited had a strong effect on age, together with genotype or age x sex interaction effects. That is, the latency of the first movement (LatM), to enter the peripheral ring (LatP), and to perform the first rearing (LatR) were strongly dependent on age (A*, *p* = 0.031; A***, *p* = 0.001 and A***, *p* < 0.001, respectively). The performance of the first vertical exploration in the periphery was also modulated by genotype (LatR, G***, *p* < 0.001) and age x sex interaction in almost all the variables studied (A × S, *p* < 0.05). This effect was also observed in the retest, with higher latencies observed in 3xTg-AD mice and older animals in general (G and A; *p* < 0.05).

The time course (Figure 2B,C) and the total activity exhibited in the open-field test on Day 1 indicated that the main effects were attributed to age factors (A***, *p* < 0.001) in both horizontal (Figure 3A) and vertical (Figure 3B) activities, with a reduction specially observed at 16 months. Regarding genotype differences, 3xTg-AD males presented decreased levels of horizontal activity, while in females, these differences were more detectable when evaluating vertical activity (G*, *p* < 0.05). The horizontal activity was further examined through gait analysis, as illustrated in Figure 3C,D. The evaluation of locomotion was measured by the pauses performed during the test and the mean distance covered in each walk. The mean walking distance of NTg mice was about three–four crossings, the number the mice needed to cover to move from one corner to the other in the peripheral and more protected areas of the test. This performance also showed genotype and age effects. The reduced activity on 3xTg-AD mice was also related to fewer crossings per unit of movement (G**, *p* < 0.01).

When evaluating the recognition task of animals, the object recognition test (Figure 3E,F) showed a low sensitivity to the genotype (as most animals were unsuccessful in reaching the acquisition criteria in 600 s) and was only noticed in male subgroups (G*, *p* = 0.040). An age effect was observed in the sample trial, with 16-month-old animals presenting with a lower time exploring the objects (A*, *p* = 0.023). No differences in the preference for the new object were observed in the test trial.

In the marble test (Figure 4A), the most sensible variable was the number of marbles completely buried. An age-dependent progressive increase of buried marbles was observed until 12 months of age. At 16 months, in agreement with the decreased activity observed at this age, a lower number of buried marbles was observed (A***, *p* = 0.000). In females, the genotype effect was represented by a higher number of buried marbles presented in 3xTg-AD female mice in comparison to NTg (G*, *p* = 0.027).

In the T-maze (Figure 4), when measuring the spontaneous alternation, an age effect was observed, with older animals taking a long time to reach the intersection of the vertical arm (Figure 4B; A*, *p* < 0.05). The analysis of errors revisiting the arms that had been already explored indicated a significant genotype and age effect (G*, A**, *p* < 0.035) with an increased number of errors of 3xTg-AD in comparison with NTg mice and in older animals until 12 months of age (Figure 4C).

In the test trial for the working memory paradigm, the mice that completed 20 s in the forced arm were evaluated, and the genotype effect was detected, especially in male mice who presented longer latencies in the test (G*, *p* < 0.05). Although no significant differences were observed when evaluating the number of errors, 3xTg-AD mice spent more time to reach the correct arm during the free choice sessions (Figure 4D, G**, *p* = 0.006).

Figure 5 illustrates the performance of animals in three paradigms of the Morris water maze, namely the cue, place task, and probe trial. Sex differences were found in the first CUE task for visual perceptual learning when the distance was studied (S*, *p* = 0.015), with females presenting higher distances taken to arrive at the platform. Day-by-day analysis of the mice’s performance in the maze showed distinct behaviors depending on the genotype and age, especially from the third day of the place task trial (G*, A*, *p* < 0.05). Thus, the time acquisition curves of both genotypes differed, with a worse performance in 3xTg-AD mice as compared to the NTg littermates, demonstrated by a longer delay in finding the platform on the third (PT3; G**, *p* = 0.008) and fourth (PT4; G* *p* = 0.01) days of the test. Genotype and age differences in swimming speed were also observed (G*, A*, *p* < 0.05) with a slower pattern in 3xTg-AD and 16-month-old mice. Therefore, distances covered to reach the platform were also calculated (Figure 5B), where an age discrepancy was observed as 16-month-old animals presented with decreased time and distance to arrive. In the probe trial (Figure 5D,F), the preference of NTg for the target quadrant was significantly higher compared to 3xTg-AD mice, as they showed an increased distance performed in the target quadrant (especially in males). Moreover, NTg showed a better latency time to reach the platform (G*, *p* < 0.05).

Age and sex differences were found in body weight (Figure 6A). An increasing tendency was observed in aged animals until 12 months (A***, *p* < 0.000). Males also presented with an increased body weight at all ages (S***, *p* < 0.000). Frailty score (Figure 6B) was higher in 3xTg-AD and aged mice (G**, A***, *p* < 0.002); sex differences were also observed, with slightly increased scores in males (S*, *p* = 0.049).

### 2.2. Isolation May Modify Hyperactivity and Neophobia Patterns and Disrupt the Obsessive Compulsive Disorder-like Digging Ethogram

Figure 7 illustrates the impact of social isolation on the behavioral and functional phenotype of 16-month-old male 3xTg-AD mice compared to a grouped-housed age mimicking the advanced stage of the disease. Different variables assessed in the open-field test are represented in Figure 7A; the time course of the horizontal activity indicated that isolated 3xTg-AD mice exhibited a hyperactive pattern performing a higher number of crossings, especially in minute 4 (*t*-test; * *p* = 0.017). Moreover, increased neophobia was also observed when comparing the number of crossings performed in the first minute of the repeated test, with a burst of initial locomotor performance observed in 3xTg-AD isolated animals (*t*-test; * *p* = 0.017).

In the MB, isolation broke the habitual digging ethogram with an increased number of buried marbles compared to the group-housed animals (Figure 7B, *t*-test; * *p* = 0.029). Although isolated animals showed lower body weight and higher frailty index, no statistical differences were observed.

### 2.3. Age-, Sex-, and Region-Dependent Accumulation of Intraneuronal Aβ, Amyloid Plaques, and Aggregated Tau in 3xTg-AD Mice

To assess the age and sex-dependent accumulation of Aβ pathology in male and female 3xTg-AD mice, immunostaining procedures were performed in different areas related to AD from the asymptomatic, prodromal, onset, advanced, and very advanced stages of the disease (2-, 4-, 6-, 12-, and 16-month-old mice, respectively). Sections corresponding approximately to bregma +1.5 mm (PFCx), −1.5 mm (AMG and HCd), and −3 mm (HCv and ECx) were obtained with immunostaining with an Aβ42-specific antibody (n = 4/sex/age). As expected, no signal was observed in NTg control mice. Intracellular signals in all areas studied were presented in 100% of males and 75% of females of 2-month-old mice and 100% of 4-month-old females. By 16 months of age, however, no Aβ42 immunoreactivity was presented in one of four males and females in the dorsal hippocampus and entorhinal cortex. On the other hand, extracellular Aβ plaques were absent in the first stages of the disease. They were first detected in females of 12-month-old mice and in 16-month-old males, firstly in the ventral hippocampus (100% of 12-month-old females and 75% of 16-month-old male) and secondarily in the entorhinal cortex (50% of 12-month-old female and 50% of 16-month-old male)” (Table 2). Notably, the subiculum was the hippocampus area with mostly extracellular Aβ plaques.

Semi-quantitative analysis considering the percentage of intracellular positive Aβ signal was evaluated and estimated with ImageJ (Figure 8). In this case, age-dependent progression of intracellular Aβ immunoreactivity was observed in PFCx, predominantly in layer V (A*, *p* = 0.045). Moreover, 3xTg-AD females presented with a higher percentage of intracellular cells in PFCx (S*, *p* = 0.043) and CxE (S**, *p* = 0.010) compared to 3xTg-AD males. Regarding brain areas, the percentage of positive cells was significantly increased in the amygdala and ventral hippocampus (Bonferroni post hoc test, *p* = 0.000).

To study the age- and sex-related tau phosphorylation in 3xTg-AD mice, sections from 2-, 4-, 6-, 12-, and 16-month-old mice were immunostained with antibodies that recognize hyperphosphorylated tau at pSer202/pThr205 (AT8). As expected, in NTg mice, tau expression was not observed. In contrast, AT8 immunoreactivity was present from the asymptomatic stages of the disease. In this case, immunoreactivity was observed throughout the ventral hippocampus and amygdala in 100% of the mice and in the entorhinal cortex in 75% of males and 100% of females. At 16 months, all males presented with AT8-positive cells in all areas studied. However, in females, this percentage decreased compared to younger 6- and 12-month-old mice (Table 2).

Semi-quantitative analysis (Figure 8) demonstrated age-related differences in the dorsal hippocampus of 3xTg-AD mice (A**, *p* = 0.01). Sex differences were evident in the amygdala (S*, *p* = 0.035). Concretely, at 6 months, these sex differences were observed in the amygdala, ventral hippocampus, and entorhinal cortex, with females exhibiting a more significant percentage of tau-positive cells than males (*p* < 0.035). Regarding different brain areas using one-way ANOVA analysis with Bonferroni’s correction, we found that immunoreactivity was higher in the amygdala and ventral hippocampus (Bonferroni post hoc test, *p* < 0.007). The entorhinal cortex also presented an increased percentage of positive cells in comparison with the prefrontal cortex (Bonferroni post hoc test, *p* = 0.018).

Together, these results indicate that, in general, the ventral hippocampus and entorhinal cortex were the most affected areas regarding amyloid plaques, and the amygdala and ventral hippocampus regarding intracellular Aβ immunoreactivity and pSer202/pThr205 tau phosphorylation. Moreover, heterogeneity and survival paradigms were detected, especially in 16-month-old females.

## 3. Discussion

### 3.1. Animal Models Are Invaluable Tools for Studying Mechanisms of AD Pathogenesis

Animal model characterization can help us to better understand AD disease mechanisms and pathogenesis. The shorter lifespan of most animal models provides a fleet-footed scenario for the study and long-term monitoring of factors potentially involved in disease modulation at the morphological, structural, functional, and behavioral levels.

Although there are very valid animal models that reproduce the neuropathology and behavioral deficits associated with dementia, none of the existing models fully represent the full spectrum of this insidious human disease [38]. Genetically modified rodent models became a reality in the mid-1990s with the PDAPP model [39]. Since then, several models based on APP, PS1, and tau mutations have been described. They are used to better understand the characteristic behaviors and neuropathological mechanisms of sporadic AD, although some differences have been found between familial and sporadic forms [40,41]. The 3xTg-AD mice are based on the familial AD mutations PS1M146V and APPSwe, which also harbor the human transgene tauP301L. They progressively develop time- and region-specific development of amyloid β-plaques and tau-containing neurofibrillary tangles and exhibit cognitive and behavioral symptoms such as those found in AD patients [30,31,32].

The complex process of aging results in old age being the most heterogeneous period of life [42]. Regarding AD patients, the heterogeneous and complex clinical profile of individuals mainly described at end-of-life dementia requires more precise diagnostic criteria to identify relatively homogeneous patient populations.

At the translational level, experimental gerontologists emphasize the relevance of using aged animals to mimic humans’ complexity and multifactorial aging processes [25,26]. However, there is a need for more literature comparing animals from very young to advanced ages due to the smaller sample size of survival-aged animals, the associated higher laboratory costs, and the complex heterogeneity of the age-related scenario. A better understanding of the interactions between age, sex, and AD phenotypic traits in animal models may contribute to developing and implementing a precision medicine approach [43].

In the present work, we determined the age, AD genotype, and sex sensitivity of a battery of behavioral tests in 3xTg-AD male and female mice at asymptomatic (2 months), prodromal (4 months), onset (6 months), advanced (12 months), and very old (16 months) ages and compared them with age-matched non-transgenic mice with normal aging. We also examined the temporal and spatial progression of Aβ pathology and tau hyperphosphorylation and studied the sex-dependent differences.

### 3.2. Age, Genotype, and Sex Modulation in Behavioral Signatures for Emotional, Cognitive, and Physical Phenotypes

The behavioral profiles of male and female 3xTg-AD and NTg mice with normal aging were assessed through seven tests that determine the effects of genotype, sex, and aging process on their motor, emotional, and cognitive functions, as well as their daily life activities. Generally, aging was shown to be the most determinant factor for most behavioral variables studied. In contrast, the genotype factor was specific to those variables related to horizontal and vertical activities, thigmotaxis, coping with stress strategies, working memory, and frailty index. Sex effect was predominantly observed in a classical emotional variable and physical status, but also the horizontal and vertical activity in the test of neophobia and the open field. These data agree with previous results obtained by our laboratory with 3xTg-AD and APP23 mice. There, the age factor was also a determinant of variables related to sensory and motor functions, whereas genotype differences were specifically indicative of cognitive and BPSD hallmarks of the disease [44,45]. According to life expectancy, the age factor overshadowed genetic differences, and mortality bias was also found [27].

It has been repeatedly reported that fear and anxiety-like behaviors are the most common NPS alterations associated with dementia that can be studied in the 3xTg-AD model of AD [35]. For this reason, the battery of tests started with the corner and open-field tests to evaluate the quantitative and qualitative features elicited in these fear-inducing enclosures. In the present work, neophobia was increased in 3xTg-AD mice, presenting a lower number of visited corners, and worsened at 16 months of age, consistent with the worsening of anxiety-like symptoms with disease progression [46]. Moreover, the sex effect was observed with females visiting more corners. These responses agreed with behaviors observed in the open-field test: aged mice showed increased latencies in all variables analyzed after direct exposure to the open and illuminated area. Genotype differences were observed in the latency to present the first rearing. When evaluating the total vertical and horizontal activity, a clear anxiogenic hypoactive pattern was observed with age-decreased activity in all variables studied, and horizontal and vertical hypoactive patterns were observed in 3xTg-AD males and females, respectively. These anxiety-like behaviors with decreased active movement and less activity have been reported in 3xTg-AD and old-age mice [35,47,48,49,50].

In the open-field test, the evaluation of locomotion, with the distinction of inactivity, allows the differentiation of hyperactivity patterns [51], and is usually measured by the walking speed. In this work, as proposed by Giménez-Llort et al. [44], gait analysis of locomotor activity was measured by the pauses performed during the test and the mean distance covered in each walk. Similar to the results obtained in the other work, the mean walking distance of NTg mice was about three–four crossings, which is the number of crossings that the mice need to cover to move from one corner to the other in the peripheral and more protected areas of the test. As in the other variables studied, age and genotype differences were observed, presenting a decrease in mean steps performed by aged and 3xTg-AD mice. Interestingly, human studies have shown abnormal gait phenotypes in the early ages of dementia which worsen in the advanced stages of dementia. Moreover, a slower fast-walking speed is a marker of frailty and mortality [52,53,54].

Burying behavior was first described in the wild as a defensive response and an anxiety state reflex of animals [55]. This behavior was also later proposed to identify biological impacts and assess screen drugs for obsessive compulsive disorder and psychotic symptoms [56,57]. A genotype effect has been observed in this laboratory with increased marble-burying in middle-aged male 3xTg-AD and was related to an anxiety-like profile [56]. This work showed a specific pattern for females with more buried marbles. At advanced stages of the disease, and in accordance with other experiments, a lower number of buried marbles were observed [28,29].

Many conditioned memory tests are based on the re-exposure to fearful contexts [36,58]. For this reason, the re-exposure to the open field and the preference for a novel object from one previously inspected were performed on the second day of tests. As previously shown in 3xTg-AD animals, the first-minute performance of a repeated open-field test is sensitive to genotype, and 3xTg-AD mice did not benefit from previous experience in the test [59].

Although the T-maze test has been mainly used to evaluate spatial working memory [60,61], coping with stress strategies and risk assessment can also be assessed with the latencies to reach the intersection of the maze. The spontaneous alternation in this black corridor of the maze resembling burrows can also evaluate the cognitive and anxiety-like profile. In this regard, the mean time to reach the intersection, a paradigm related to immunosenescence and reduced survival [62], was increased with age. Genotype and age increased the number of errors with 3xTg-AD and older animals revisiting the arms that had already been explored. However, and in accordance with other studies, a convergence of behavioral profiles was observed at 16 months [27,29]. When evaluating the errors attributed to working memory, 3xTg-AD mice spent more time choosing the right arm in the free-choice session, with males significantly affected. These findings support the results of Setevens et al., who found working memory deficits in a radial maze in 3xTg-AD between 2 and 15 months [63].

In the Morris Water maze, short- and long-term spatial reference learning and memory were assessed [64]. Swimming speed was slower in 3xTg-AD and older mice. For this reason, means of escape latency and distance covered to reach the platform were evaluated. Sex differences were found in the cued learning, with females presenting with higher distances to arrive at the platform. No other differences were observed in this regard. Day-by-day analysis showed distinct behaviors with a worse performance in 3xTg-AD mice and older mice; however, age discrepancy was observed, with 16-month-old animals presenting performances which were not as bad as expected, with decreased times and distances taken to arrive to the platform. These results could be explained due to mortality bias (with the death of the animals which performed worse) and the age improvement of both 3xTg-AD and NTg groups [44]. As expected, in the probe trial, the preference of NTg (especially in males) for the target quadrant was significantly higher compared to 3xTg-AD mice. Moreover, they showed better latency time to reach the platform.

Heterogeneity is found in aging, and prognostic tools to identify end-of-life dementia stages are difficult [4]. The frailty concept has become a standard tool for measuring human health status and a comprehensive tool for predicting disease outcomes and mortality [65]. This tool is relatively new in animal research, and different approaches to defining frailty status have been described and proposed as predictors of an animal’s lifespan [66,67,68]. Moreover, frailty data in animal models of neurodegenerative disorders are limited. In our case, frailty parameters were evaluated using the Mouse Clinical Frailty Index, a translational adaptation [69]. As described previously, in terms of the most common frailty clinical presentations in aged mice, the integumentary and muscular–skeletal systems were the most usual variables affected [28,70]. Frailty scores increased with age, and 3xTg-AD mice presented higher scores than non-transgenic mice. These data will agree with other studies performed with the same frailty index tool in 3xTg-AD and 5xFAD transgenic mice [71,72]. Moreover, and in contrast with the data observed in the general population, with women usually presenting higher scores [73], sex differences were also observed with slightly increased scores in males.

### 3.3. Effects of Social Housing Conditions on Behavioral Signatures

Rodents are social animals; for this reason, housing conditions can modify animals’ behavior, as social interaction is essential for their welfare. It has been described that single isolated conditions may interfere with behavioral and physiological parameters [74]. In humans, loneliness has also been related to functional decline and health morbidities [75,76]; in AD, social isolation is associated with increased memory decline and aggravation of neuropsychiatric symptoms [77,78]. Due to increased male mortality and disruptive behaviors, many 3xTg-AD mice arrive at old age in socially isolated housing conditions. In this study, some animals arrive at 16 months in a long-term naturally occurring isolation scenario. Considering the extreme difficulties in obtaining a correct sample size of ancient animals, we performed a secondary analysis with these isolated animals that cannot be included in the experimental battery to measure the impact of social isolation on behavioral outcomes. The data obtained corroborate the effects described in previous works, with isolated 3xTg-AD animals presenting with a hyperactive pattern and increased neophobia behavior in the open-field test. Moreover, in the marble test, the isolation condition broke the habitual digging ethogram and increased the burying behavior [29,79,80]. Considering all these data, isolated animals should not be included in experimental groups as this can produce confounding factors and false negative behavioral outcomes.

### 3.4. Importance of Brain Regions Underlying Neuropsychiatric Symptoms in the 3xTg-AD Mouse Model

The 3xTg-AD animal model progressively develops temporal- and regional-specific development of amyloid β-plaques and tau-containing neurofibrillary tangles observed in the brains of human AD patients. In the initial reports of the 3xTg-AD model, mice first develop intraneuronal Aβ at 3–4 months of age, followed by plaque formation at 6 months of age in the cortex and hippocampus, with NFT becoming apparent at 12 months of age [30,31]. However, in recent years, several studies have indicated a drift in the phenotype of mice, with males being particularly affected [32,37].

Although NPS have been repeatedly reported in the 3xTg-AD mouse model [35,47], the relationship between NPS and the pathological mechanism of AD remains unclear. In humans, emotional and anxiety behaviors presented in AD have been associated with metabolic and volumetrics alterations in the amygdala [19,20]. In 3xTg-AD mice, although intracellular Aβ accumulation in the amygdala has been reported [33,36], the last updated characterization of the model has been focused on the hippocampus and neocortex [32,37]; taking particular account of cognitive function at the behavioral level.

On the other hand, the literature has shown that different hippocampus subregions are involved in different functions. While the dorsal hippocampus (posterior hippocampus in humans) mainly performs cognitive functions, including learning and memory, the ventral hippocampus (the anterior hippocampus in humans) has been related to emotion and stress responses. Moreover, the dorsal hippocampus relates to cortical regions involved in information processing; the ventral hippocampus presents brain connections to the prefrontal cortex, amygdala, and hypothalamus; structures related to anxiety and fear responses, reward, and motivation [24]. Neuroimaging studies have demonstrated that hippocampal subregions presented different atrophy progression in AD patients [81].

In this way, Belfiore et al. observed that Aβ plaques and tau pathology initially appear in the caudal hippocampus and progress to the rostral hippocampus with age [32]. In our case, intracellular Aβ signals were present in all areas studied (prefrontal cortex, amygdala, dorsal and ventral hippocampus, and entorhinal cortex) from the early stages of the disease. Regarding semi-quantitative analysis, age-dependent progression was observed in layer V of the prefrontal cortex. Females presented with a higher percentage of positive cells; the amygdala and ventral hippocampus were the most affected areas. Extracellular Aβ plaques were first detected in 12-month-old females and in 16-month-old males, with the ventral hippocampus and the entorhinal cortex being the most and second-most affected areas. AT8 immunoreactivity was also present as early as two months of age throughout the ventral hippocampus and amygdala in 100% of the mice and in the entorhinal cortex of 75% of males and 100% of females. These data agree with Mufson et al., who observed that 6E10 and AT8 immunoreactivity occurs in 3-week-old animals [82].

As commented, regarding sex differences, our findings are in line with prior investigations showing females with earlier and more severe neuropathology [37]. Finally, as observed in behavior performance at late stages of the disease, increased heterogeneity and survival paradigm were detected, especially in 16-month-old females.

### 3.5. Limitations

One of the limitations of the present study is the difficulties in obtaining a correct sample size of aged animals due to the higher heterogeneity in physiological and behavioral heterogeneity presented in pathological and non-pathological aging processes. Interlaboratory reliability would be interesting to study as the generation of sublines with different onsets and progressions of symptoms have been described, even between littermates. On the other hand, a more accurate histological study could be interesting if able to demonstrate an increased number of sections per tissue analyzed to quantify the neuropathology damage more effectively, in addition to other more quantitative techniques.

## 4. Materials and Methods

### 4.1. Animals

A total of 172 male and female mice from the Spanish colonies of homozygous triple-transgenic (3xTg-AD) mice harboring human PS1/M146V, APPSwe, and tauP301L transgenes (n = 88) and non-transgenic (NTg, n = 84) mice in a C57BL/6 background were used. The 3xTg-AD mice were genetically engineered at the University of California, Irvine, as described previously [31].

All the animals were housed three to three–four per cage and maintained (Makrolon, 35 × 35 × 25 cm^3^) under standard laboratory conditions (12 h light/dark, cycle starting at 8:00 h, food and water available ad libitum, 22 ± 2 °C, 50–60% humidity). Behavioral assessments [35] were performed from 9:00 h to 13:00 h in a counterbalanced manner per genotype, sex, and (when possible) per age, blind to the experiment.

All procedures followed Spanish legislation on ‘Protection of Animals Used for Experimental and Other Scientific Purposes’ and the EU Council directive (2010/63/EU) on this subject. The protocol CEEAH 3588/DMAH 9452 was approved by Departament de Medi Ambient i Habitatge, Generalitat de Catalunya. The study complies with the ARRIVE guidelines developed by the NC3Rs and aims to reduce the number of animals used [83].

### 4.2. Behavioral Assessments

Ten sets of animals of 2, 4, 6, 12, and 16 months of age (n = 14–21 mice per each genotype and age experimental group, half of them females, with a maximum of two males and females from each litter) were successively assessed using a battery of 7 tests to evaluate four behavioral and functional dimensions: sensorimotor and cognitive (dys)functions, emotionality, NPS-like symptoms, and daily life activities.

In addition, a long-term, naturally occurring isolation scenario (9.1 ± 0.7 months) was found in twelve 16-month-old males 3xTg-AD after living in a standard social environment. Still, all the animals in the group- or under isolated-housing conditions, were socially connected through olfaction and audition, and the cages were enriched with nesting materials. Therefore, a secondary analysis was performed to measure the impact of social isolation on behavioral outcomes and physical status in these animals using the same battery of tests except for the MWM. 

#### 4.2.1. Day 1: Corner and Open-Field Tests (CT and OF)

Animals were individually placed in the center of a clean standard home cage filled with wood-shaved bedding. Neophobia was evaluated in the corner test (CT) for 30 s by measuring the number of corners visited, latency to perform the first rearing, and the number of rearings. Immediately after, mice were placed in the center of an open-field (OF) beige metal drawer (metalwork, 42 × 38 × 15 cm^3^) and were observed for 5 min. The sequence of behavioral events that defined the animals’ ethogram was recorded as follows: duration of freezing behavior (latency to move), latency to leave the central square and enter the peripheral ring, latency, and total duration of self-grooming behavior. Horizontal (crossings of 10 × 10 cm^2^ squares) and vertical (rearings with wall support) locomotor activities were also measured. During the tests, defecation boli and urination were also recorded.

Gait analysis of the locomotion in the open field was used to measure the number of forward locomotion episodes (walking preceded and followed by a rest) and the number of crossings covered on each, as complementary to measures of horizontal and vertical activity time courses [35] and classical total counts [44]

#### 4.2.2. Day 2: Recognition Tests (OF2 and OR)

The animals were retested in the open field (OF2) the day after to evaluate their behavioral response when they again confronted the same anxiogenic environment. Activity analysis was carried out during the first minute of the test, the period where we have previously described AD genotype differences [59]. Immediately after, animals were moved to a standard home cage, where they remained for one minute before being reintroduced to the field, where two objects were now allocated to administer the novel object recognition test. The animals were assessed for their ability to recognize a familiar object (S, sample) from a new one (N). The animals were placed in the open field (a known environment) in the sample trial. They left to explore (nose directed to the object not less than 1 cm) two identical objects, S1 and S2 (glass bottles, 15 × 12 cm, 5 cm diameter), equally spaced on the floor of the apparatus until they reached the criteria of exploration of both for 20 s until a maximum time of 600 s. Two hours and a half later, animals were reintroduced to the apparatus for 5 min (test trial), where two different non-explored objects were located: an identical copy of the sample objects (S3) and a completely new object (N, rectangular aluminum can, 15 × 10 cm, 4 cm high). Preference for the new object was measured through the index TN − TS/TN + TS, where TS and TN are the time spent exploring “S3” and “N”, respectively.

#### 4.2.3. Day 3: Marble Test (MB)

Mice were placed individually in a standard home cage containing nine glass marbles (dimensions 1 × 1 × 1 cm^3^) evenly spaced, making a square (three rows of three marbles per row only in one-quarter of the cage) on a 5 cm thick layer of sawdust [56]. The mice were introduced in the zone without marbles facing the wall and left to interact with the cage freely. After 30 min, the mice were gently removed from the cage, and the level of marbles’ burying was measured: intact (untouched), rotated (90° or 180°), half-buried (at least ½ buried by sawdust), and buried (completely hidden).

#### 4.2.4. Days 4 and 5: Spontaneous Alternation and Working Memory Paradigm in the T-Maze (TM-SA and TM-WM)

Two different paradigms were carried out in a T-shaped maze (woodwork; two short arms of 30 × 10 cm^2^; one long arm of 50 × 10 cm^2^). In the spontaneous alternation task [84], coping with stress strategies, risk assessment, and working memory were assessed in a single trial. The animals were placed inside the maze’s long arm with their head facing the end wall, and they were allowed to explore the maze for a maximum of 5 min. The latencies of the first movement, arriving at the intersection of the arms, and completing the exploration of the maze were recorded [85]. The entry of an already visited arm in the trial before completing the test was considered an error. Defecation boli and urination were also noted.

Twenty-four hours later, a working memory paradigm was used. It consisted of two consecutive trials: one forced choice followed, 60 s later, by one free choice (recall trial). The latencies of the first movement, arriving at the intersection of the arms, and the time elapsed until the mice completed 20 s in the forced arm (time to reach the criteria) were recorded. Sixty seconds later, the animals that completed the forced trial in less than the cut-off time (10 min) were allowed to explore the maze in the free choice trial where both arms were accessible for 5 min. The arm the mice chose, and the time spent reaching the correct arm during the free choice, were recorded (exploration criteria). The choice of the previously visited arm in the previous trial was considered an error, and the total number was calculated. Finally, defecation boli and urination were also recorded [86].

#### 4.2.5. Days 6 to 10: Morris Water Maze (MWM)

Three paradigms assessed learning and memory in the Morris water maze (MWM) [87]. Mice were trained with four trials per day, spaced 30 min apart, to locate a platform (11 cm diameter) in a circular pool for mice (120 cm diameter, 80 cm height, 25 °C opaque water), and were covered with a completely black curtain. Mice were gently released (facing the wall) from one randomly selected starting cardinal point and allowed to swim until they escaped onto the platform.

On the first day, a cue task (CUE, DAY 1) assessed the visual perceptual learning and memory of a visible platform elevated 1 cm above the water level in the NE position and indicated by a visible striped flag (5.3 × 8.3 × 15 cm). Extra maze cues were absent in the black curtain. During the next four consecutive days (PT1-PT4, DAY2-DAY5), the mice searched for a hidden platform in the middle of the SW quadrant. Different geometric figures hung on each room wall were used as external visual clues. In all trials, mice failing to find the platform within 60 s were placed on it for 10 s, the same period as the successful animals. On the last day, 2 h and 30 min after the last trial of the place-learning task, the removal, a probe trial without the platform, was administered for 60 s to assess spatial memory for the previously trained platform location.

In all the learning tasks, the variables of time (escape latency), distance, and swimming speed were also recorded by a computerized tracking system (ANY-Maze v. 5.14, Stoelting, Dublin, Ireland). The number of crossings over the removed platform position (annulus crossings), the time spent, and the distance traveled in each quadrant were also analyzed.

### 4.3. Physical Status: Body Weight (BW) and Mouse Clinical Frailty Index Assessment (FI)

After the behavioral assessment, the body weight was recorded. Frailty was assessed using an adaptation of the MCFI [69], including 30 “clinically” assessed non-invasive items. For 29 of these items, mice were given a score of 0 if not presented, 0.5 if there was a mild deficit, and 1 for a severe deficit. Weight was scored based on the number of standard deviations from a reference mean. The clinical evaluation included the integument, the physical/musculoskeletal system, the vestibulocochlear/auditory systems, the ocular and nasal systems, the digestive system, the urogenital system, the respiratory system, signs of discomfort, and body weight.

### 4.4. Brain Samples and Immunofluorescence Analysis

For immunofluorescence analysis, 3xTg-AD mice (n = 4/sex/age, counterbalanced per litter and experimental set) were anesthetized with a mixture of ketamine/xylazine and perfused with NaCl 0.9% followed by formaldehyde 3.7%. Brains were cryoprotected in sucrose, frozen in isopentane, and finally stored at −80 °C until the realization of free-floating coronal brain sections (30 µm). NTg mice of each age and sex were evaluated as good negative controls of the technique and the antibodies. Immunostaining procedures were performed in sections of different neuroanatomical areas related to AD: the prefrontal cortex (PFCx), entorhinal cortex (ECx), amygdala (AMG), dorsal hippocampus (HCd), and ventral hippocampus (HCv). For Aβ staining, sections were treated with 60% formic acid (6 min), a protocol that allows specific labeling of Aβ over full-length APP as described [36]. For tau staining, sections were pretreated with citrate buffer and heated (30 min at 80 °C) for antigen retrieval. Sections were washed in DPBS and incubated overnight at 4 °C with antibodies against human Aβ1-16 (6E10; 1:2000; Biolegend, San Diego, CA, USA) and hyperphosphorylated tau (pSer202/pThr205: AT8; 1:1000; Thermo Fisher scientific, Waltham, MA, USA). Sections were washed to remove excess antibody and incubated in the suitable secondary antibody (Donkey anti-mouse Alexa 555 and Alexa 488; 1:500; Thermo Fisher Scientific) for two hours at room temperature, counterstained with DAPI and the slices were mounted and coversliped with aqueous mounting media. Images from three sections corresponding approximately to bregma +1.5 (PFCx), −1.5 (AMG and HCd), and −3 (HCv and ECx) were obtained using an Eclipse 90i microscope (Nikon, Melville, NY, USA) at 40×, and the percentage of positive Aβ and tau cells were evaluated and estimated using ImageJ (https://imagej.net/ij/, accessed on 28 August 2023). Representative images of each brain area were obtained with Confocal TCS SP5 (Leica, Wetzlar, Germany) using a 10× objective with a 4× digital zoom.

### 4.5. Statistics

Statistical analyses were performed using SPSS 15.0 (SPSS Inc., Chicago, IL, USA) and GraphPad Prism 8.0 (GraphPad Software Inc., San Diego, CA, USA). Results are expressed as mean ± SEM or percentage. To evaluate the effects of (G) genotype, (A) age, and (S) sex, a factorial analysis design was applied through a multivariate general lineal model analysis. One-way ANOVA was used to analyze differences in immunofluorescence analysis. Bonferroni’s multiple-comparison test and Student *t*-test comparisons were used for differences between isolated or grouped animals. In all cases, statistical significance was considered at *p* < 0.05.

## 5. Conclusions

In the present work, we determined the age, AD genotype, and sex sensitivity of behavioral tests’ battery in 3xTg-AD male and female mice at asymptomatic (2 months), prodromal (4 months), onset (6 months), advanced (12 months), and very old (16 months) ages and compared them with age-matched non-transgenic mice with normal aging. Animals were evaluated in a battery of seven behavioral tests to comprehensively screen motor, non-cognitive, and cognitive-like symptoms. On the other hand, we examined the temporal and spatial progression of Aβ pathology and tau hyperphosphorylation and studied the sex-dependent differences.

In summary, most of the variables analyzed showed age-related differences. In contrast, the genotype factor was specific to those variables related to horizontal and vertical activities, thigmotaxis, coping with stress strategies, working memory, and frailty index. Sex effect was predominantly observed in a classical emotional variable and physical status.

Non-linear age- and genotype-dependent behavioral signatures were found in 16-month-old mice, suggesting a compensation mechanism and survival bias through physiological and pathological aging. Investigating this mechanism may help better understand individual heterogeneity in the advanced stages of dementia.

On the other hand, intraneuronal Aβ pathology and tau hyperphosphorylation has been present since the first stages of the disease, placing special importance on the amygdala and ventral hippocampus. This fact makes 3xTg-AD mice a valuable model to study neuropathological mechanisms involved with neuropsychiatric symptoms related to AD, taking age and sex factor interactions into account.

## Figures and Tables

**Figure 1 ijms-24-13796-f001:**
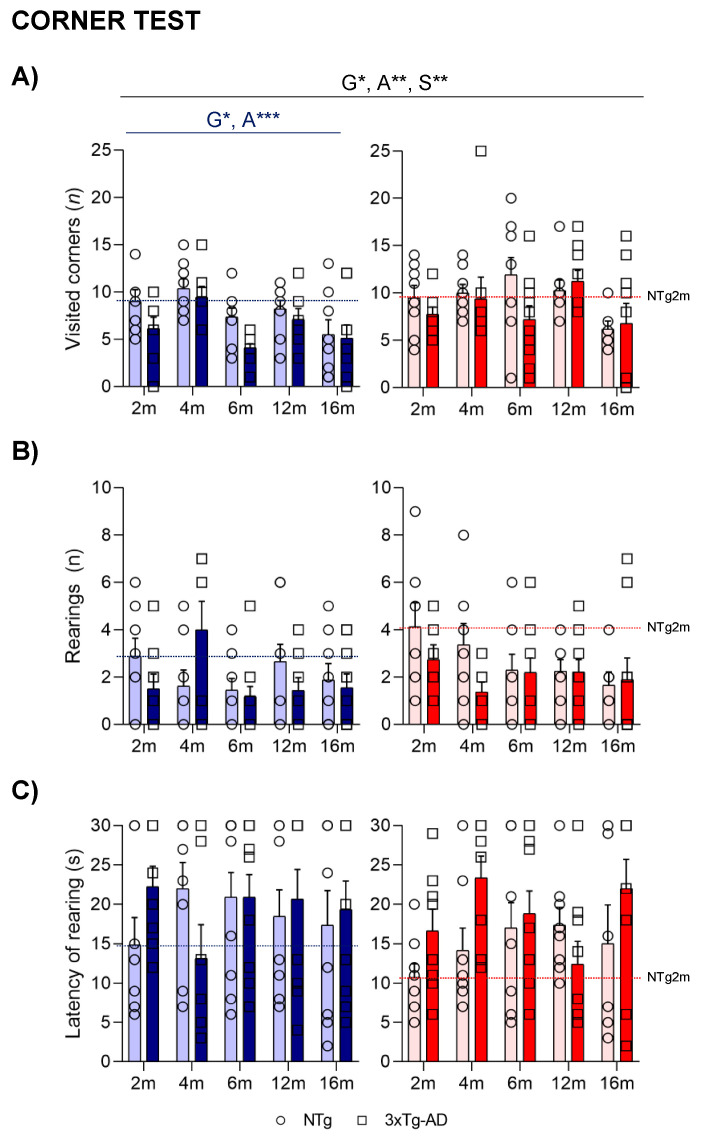
NPS-like behaviors in the corner test—Effect of AD genotype, age, and sex on the neophobia elicited in a new home cage at 2, 4, 6, 12, and 16 months. 2m-NTg (n = 16, 8 males and 8 females); 2m-3xTg-AD (n = 16, 8 males and 8 females); 4m-NTg (n = 16, 8 males and 8 females); 4m-3xTg-AD (n = 15, 7 males and 8 females); 6m-NTg (n = 21, 11 males and 10 females); 6m-3xTg-AD (n = 21, 11 males and 10 females); 12m-NTg (n = 17, 9 males and 8 females); 12m-3xTg-AD (n = 18, 9 males and 9 females); 16m-NTg (n = 14, 8 males and 6 females); 16m-3xTg-AD (n = 18, 9 males and 9 females).Visited corners (**A**), number (**B**), and latency (**C**) of rearings. Data are expressed by Mean ± SEM. Individual values: Circles: NTg mice; Squares: 3xTg-AD mice; Mean values: Blue bars: males; Red bars: females; Ice blue: NTg males; Navy blue: 3xTg-AD males; Pale red: NTg females; Carmine red: 3xTg-AD females. Pointed lines: averaged performance of NTg mice of that sex at 2 months of age is indicated as a reference level. Statistics: Analysis of variance: Statistics: G, genotype; A, age; S, sex; * *p* < 0.05, ** *p* < 0.01, *** *p* < 0.001.

**Figure 2 ijms-24-13796-f002:**
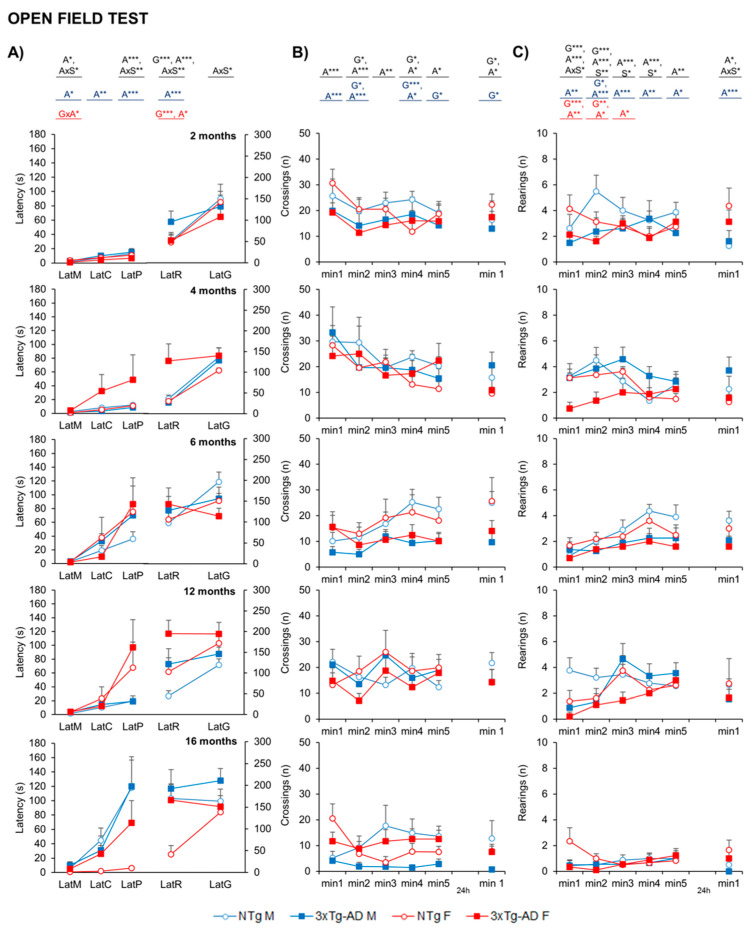
NPS-like behaviors in open-field test—Effect of AD genotype, age, and sex on the neophobia and exploratory activity elicited in the open-field test at 2, 4, 6, 12, and 16 months. 2m-NTg (n = 16, 8 males and 8 females); 2m-3xTg-AD (n = 16, 8 males and 8 females); 4m-NTg (n = 16, 8 males and 8 females); 4m-3xTg-AD (n = 15, 7 males and 8 females); 6m-NTg (n = 21, 11 males and 10 females); 6m-3xTg-AD (n = 21, 11 males and 10 females); 12m-NTg (n = 17, 9 males and 8 females); 12m-3xTg-AD (n = 18, 9 males and 9 females); 16m-NTg (n = 14, 8 males and 6 females); 16m-3xTg-AD (n = 18, 9 males and 9 females). Data are expressed by Mean ± SEM. Circles: NTg mice; Squares: 3xTg-AD mice; Mean values: Blue: males; Red: females. Ethogram (**A**), the temporal sequence of behavioral events in the open-field test, Variables: LatM, latency of the first movement; LatC, latency to leave the central area; LatP, latency to enter into the periphery; LatR, latency of the first rearing; LatG, latency of the first grooming. Time course of the horizontal activity (number of crossings) of the open-field test on Day 1: min1 to 5 and Day 2: min1 (**B**). Time course of the vertical activity (number of rearings) of the open-field test on Day 1: min1 to 5 and Day 2: min1 (**C**). Statistics: Analysis of variance: Effects of Analysis of variance: Statistics: G, genotype; A, age; S, sex; GxA, genotype x age interaction; AxS, age x sex interaction: * *p* < 0.05, ** *p* < 0.01, *** *p* < 0.001.

**Figure 3 ijms-24-13796-f003:**
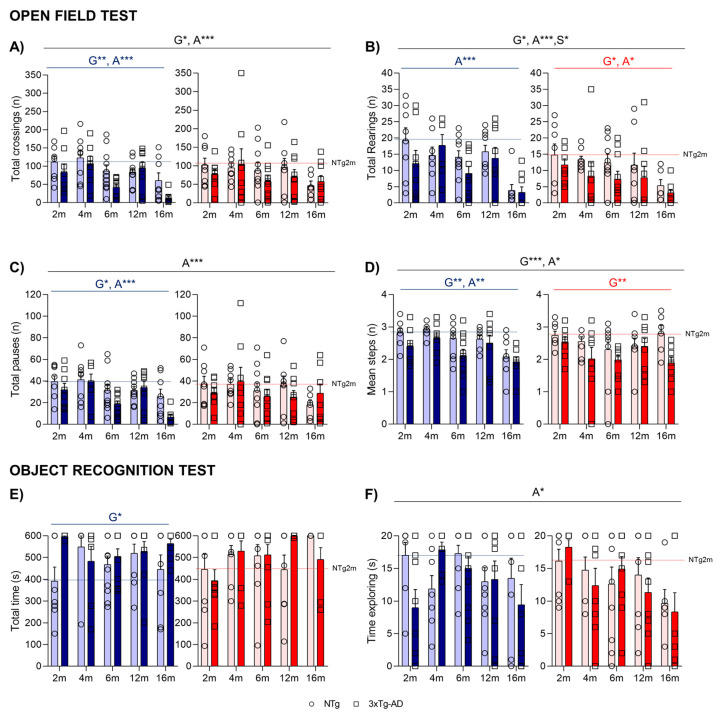
NPS-like behaviors and gait analysis in the open-field test—(**A**–**D**) and novel object recognition tasks (**E**,**F**). Effect of AD genotype, age, and sex at 2, 4, 6, 12, and 16 months. 2m-NTg (n = 16, 8 males and 8 females); 2m-3xTg-AD (n = 16, 8 males and 8 females); 4m-NTg (n = 16, 8 males and 8 females); 4m-3xTg-AD (n = 15, 7 males and 8 females); 6m-NTg (n = 21, 11 males and 10 females); 6m-3xTg-AD (n = 21, 11 males and 10 females); 12m-NTg (n = 17, 9 males and 8 females); 12m-3xTg-AD (n = 18, 9 males and 9 females); 16m-NTg (n = 14, 8 males and 6 females); 16m-3xTg-AD (n = 18, 9 males and 9 females). Total horizontal (**A**) and vertical (**B**) activity in the open-field test. Gait analysis: Total number of pauses (**C**) and mean number of crossings in the total time of the open-field test (**D**). Time to reach the 20s criteria (**E**) and total time exploring object (**F**) in the sample trial of the novel object in the object recognition test. Data are expressed by Mean ± SEM. Individual values: Circles: NTg mice; Squares: 3xTg-AD mice; Mean values: Blue bars: males; Red bars: females; Ice blue: NTg males; Navy blue: 3xTg-AD males; Pale red: NTg females; Carmine red: 3xTg-AD females. Pointed lines: averaged performance of NTg mice of that sex at 2 months of age is indicated as reference level. Statistics: Analysis of variance: Statistics: G, genotype; A, age; S, sex; * *p* < 0.05, ** *p* < 0.01, *** *p* < 0.001.

**Figure 4 ijms-24-13796-f004:**
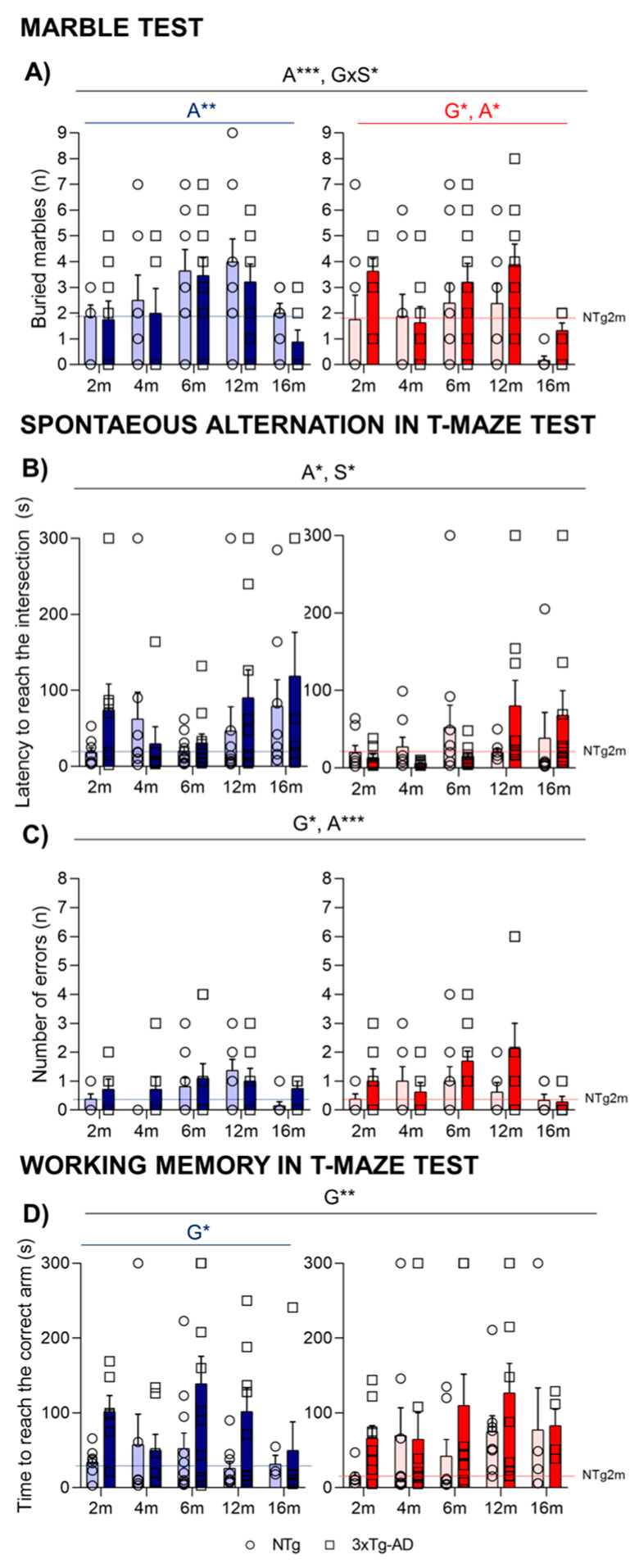
Digging behavior in the marble burying test (**A**). Coping with stress strategies, risk assessment, spontaneous alternation, and working memory in the T-Maze (**B**–**D**). Effect of AD genotype, age, and sex at 2, 4, 6, 12, and 16 months. 2m-NTg (n = 16, 8 males and 8 females); 2m-3xTg-AD (n = 16, 8 males and 8 females); 4m-NTg (n = 16, 8 males and 8 females); 4m-3xTg-AD (n = 15, 7 males and 8 females); 6m-NTg (n = 21, 11 males and 10 females); 6m-3xTg-AD (n = 21, 11 males and 10 females); 12m-NTg (n = 17, 9 males and 8 females); 12m-3xTg-AD (n = 18, 9 males and 9 females); 16m-NTg (n = 14, 8 males and 6 females); 16m-3xTg-AD (n = 18, 9 males and 9 females). Spontaneous alternation in T-Maze, latency to cross the intersection (**B**); total number of errors performed during the exploration of the maze (**C**); working memory in T-maze test; time spent to reach the correct arm (**D**). Data are expressed by Mean ± SEM. Individual values: Circles: NTg mice; Squares: 3xTg-AD mice; Mean values: Blue bars: males; Red bars: females; Ice blue: NTg males; Navy blue: 3xTg-AD males; Pale red: NTg females; Carmine red: 3xTg-AD females. Pointed lines: averaged performance of NTg mice of that sex at 2 months of age is indicated as reference level. Statistics: Analysis of variance: Statistics: G, genotype; A, age; S, sex; * *p* < 0.05, ** *p* < 0.01, *** *p* < 0.001.

**Figure 5 ijms-24-13796-f005:**
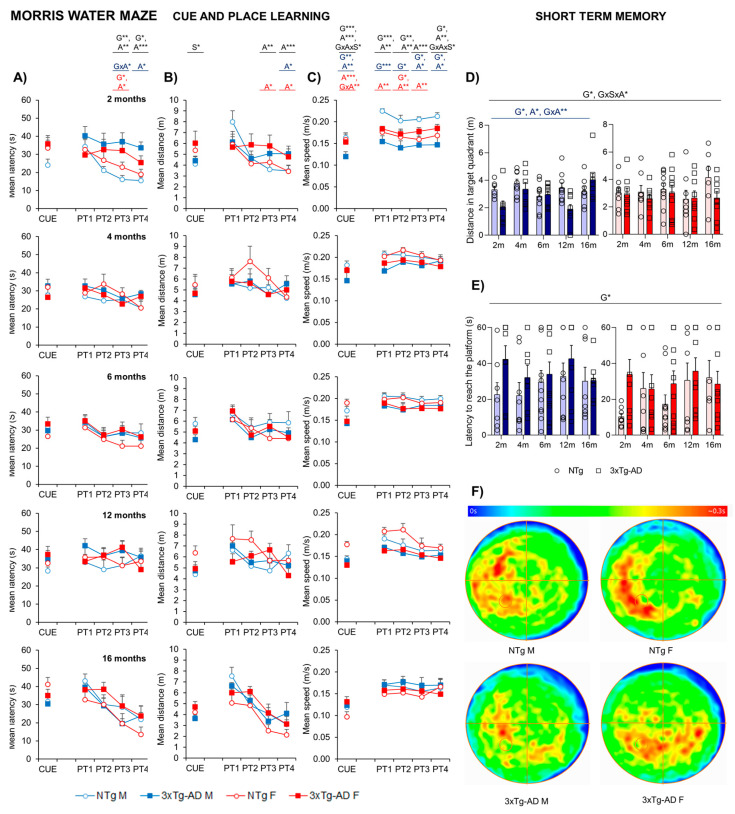
Quantitative analysis of NTg and 3xTg-AD mice performance at 2, 4, 6, 12, and 16 months in three different paradigms of the MWM. 2m-NTg (n = 16, 8 males and 8 females); 2m-3xTg-AD (n = 16, 8 males and 8 females); 4m-NTg (n = 16, 8 males and 8 females); 4m-3xTg-AD (n = 15, 7 males and 8 females); 6m-NTg (n = 21, 11 males and 10 females); 6m-3xTg-AD (n = 21, 11 males and 10 females); 12m-NTg (n = 17, 9 males and 8 females); 12m-3xTg-AD (n = 18, 9 males and 9 females); 16m-NTg (n = 14, 8 males and 6 females); 16m-3xTg-AD (n = 18, 9 males and 9 females). Data are expressed by Mean SEM. Day-by-day quantitative analysis of the CUE and place learning task (PT) by means of escape latency (**A**), distance (**B**), and swimming speed (**C**). Probe trial for short-term memory assessed by mean distance in the target quadrant (**D**) and latency to reach the platform (**E**); heat map representation of mobility in the probe trial (**F**). Data are expressed by Mean ± SEM. Individual values: Circles: NTg mice; Squares: 3xTg-AD mice; Mean values: Blue bars: males; Red bars: females; Ice blue: NTg males; Navy blue: 3xTg-AD males; Pale red: NTg females; Carmine red: 3xTg-AD females. Pointed lines: averaged performance of NTg mice of that sex at 2 months of age is indicated as reference level. Statistics: Analysis of variance: Statistics: G, genotype; A, age; S, sex; GxA, genotype x age interaction; GxAxS, genotype x age x sex interaction * *p* < 0.05, ** *p* < 0.01, *** *p* < 0.001.

**Figure 6 ijms-24-13796-f006:**
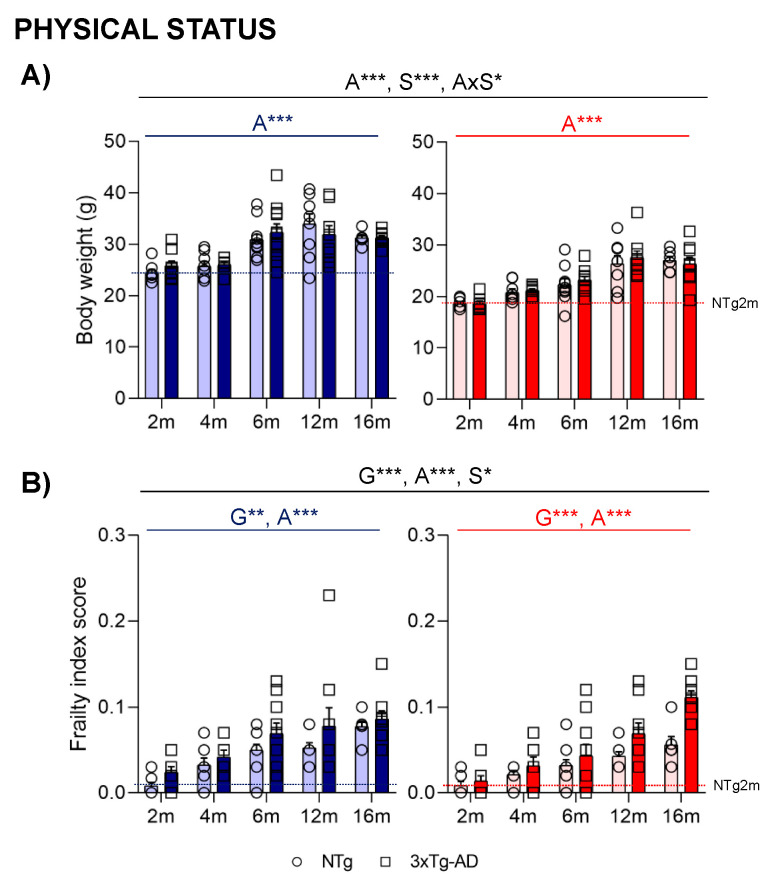
Physical status: body weight (**A**) and Mouse Clinical Frailty Index Assessment (**B**). Effect of AD genotype, age, and sex at 2, 4, 6, 12, and 16 months. 2m-NTg (n = 16, 8 males and 8 females); 2m-3xTg-AD (n = 16, 8 males and 8 females); 4m-NTg (n = 16, 8 males and 8 females); 4m-3xTg-AD (n = 15, 7 males and 8 females); 6m-NTg (n = 21, 11 males and 10 females); 6m-3xTg-AD (n = 21, 11 males and 10 females); 12m-NTg (n = 17, 9 males and 8 females); 12m-3xTg-AD (n = 18, 9 males and 9 females); 16m-NTg (n = 14, 8 males and 6 females); 16m-3xTg-AD (n = 18, 9 males and 9 females). Data are expressed by Mean ± SEM. Individual values: Circles: NTg mice; Squares: 3xTg-AD mice; Mean values: Blue bars: males; Red bars: females; Ice blue: NTg males; Navy blue: 3xTg-AD males; Pale red: NTg females; Carmine red: 3xTg-AD females. Pointed lines: averaged performance of NTg mice of that sex at 2 months of age is indicated as reference level. Statistics: Analysis of variance: Statistics: G, genotype; A, age; S, sex; AxS, age x sex interaction * *p* < 0.05, ** *p* < 0.01, *** *p* < 0.001.

**Figure 7 ijms-24-13796-f007:**
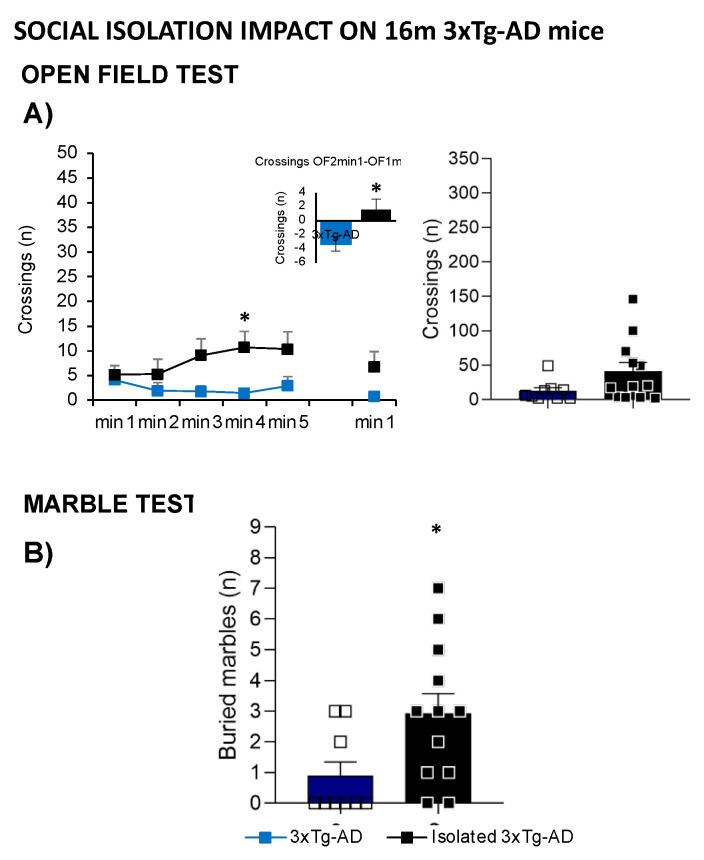
Effects of isolation on 16-month-old male 3xTg-AD mice in the open-field test (**A**) and marble test (**B**); 16m-NTg (n = 8 males), grouped-16m-3xTg-AD (n = 9 males), isolated-16m-3xTg-AD (n = 12 males). Data are expressed by Mean ± SEM.; Squares: 3xTg-AD mice; Blue: group-housed 3xTg-AD mice, black: isolated 3xTg-AD mice. Statistics: Student *t*-test comparisons vs. group-housed 3xTg-AD group. * *p* < 0.05.

**Figure 8 ijms-24-13796-f008:**
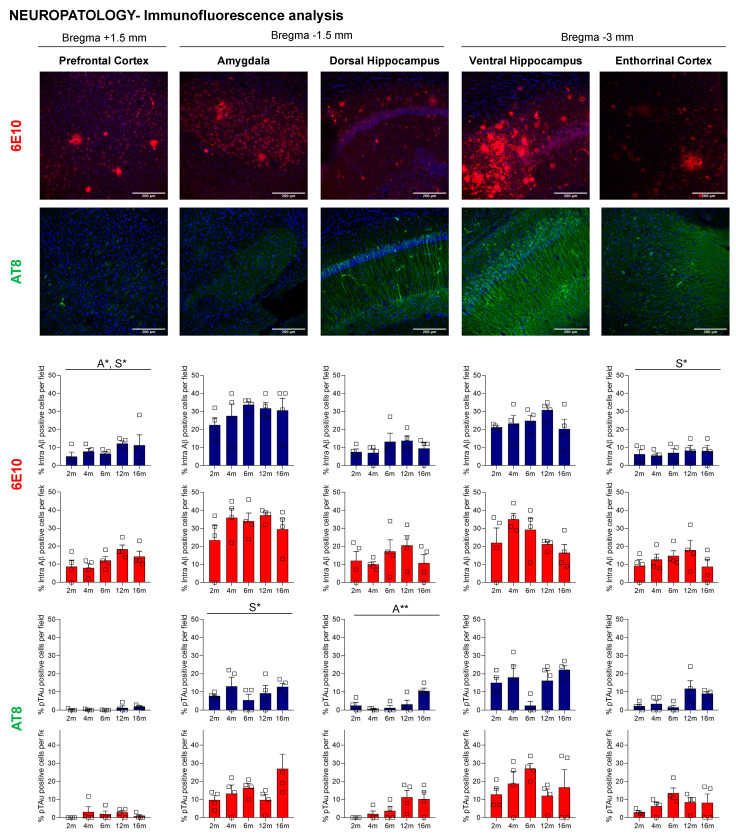
Age-dependent progression of intra Aβ-positive cells and tau phosphorylation in the prefrontal cortex, amygdala, dorsal and ventral hippocampus, and entorhinal cortex of 3xTg-AD mice. Representative confocal microphotographs of sections from 2-, 4-, 6-, 12-, and 16-month-old 3xTg-AD mice stained with 6E10 and AT8 antibodies (n = 4/age and sex group). Brain sections were selected at +1.50, −1.50, and −3.00 mm posterior to bregma. Data are expressed by Mean ± SEM. Individual values: Squares: 3xTg-AD mice; Mean values: Blue bars: males; Red bars: females; Statistics: Analysis of variance: Statistics: A, age; S, sex; * *p* < 0.05, ** *p* < 0.01.

**Table 1 ijms-24-13796-t001:** Genotype, age, and sex factors and interaction effects on behavioral tests and variables from 2 (asymptomatic) to 16 months of age (advanced stages of disease). Statistics: Multivariate general lineal model analysis. G, genotype; A, age; S, sex. * *p* < 0.05, ** *p* < 0.01, *** *p* < 0.001.

Behavioral Tests and Variables	G	A	S	G × A	G × S	A × S	G × A × S
Corner test (CT)							
Total visited corners (n)	*	**	**	-	-	-	-
Total numbers of rearings (n)	-	-	-	-	-	-	*
Latency of rearing (s)	-	-	-	-	-	-	-
Open-field test (OF)							
Freezing—Latency of first movement (s)	-	*	-	-	-	*	-
Latency to exit the center (s)	-	-	-	-	-	-	-
Latency to entering the peripheral ring (s)	-	***	-	-	-	**	-
Latency of rearing (s)	***	***	-	-	-	**	-
Latency of self-grooming (s)	-	-	-	-	-	*	-
Total horizontal activity (n crossings)	*	***	-	-	-	-	-
Total horizontal activity in the center (n)	-	*	*	-	-	-	-
Total horizontal activity in the periphery (n)	*	***	-	-	-	-	-
Total vertical activity (n rearings)	*	***	*	-	-	-	-
Gait analysis—Total number of pauses (n)	-	***	-	-	-	-	-
Gait analysis—Mean number of crossings (n)	***	*	-	-	-	-	-
Defecation (n)	*	-	-	-	-	-	-
Urination (%)	-	-	***	-	-	-	-
Context and object recognition tests							
OF2—Freezing—Latency of first movement (s)	-	***	-	-	-	*	-
OF2—Latency to exit the center (s)	***	***	-	-	-	*	-
OF2—Latency to entering the peripheral ring(s)	***	***	-	-	-	-	-
OF2—Latency of rearing (s)	*	**	-	-	-	*	-
OF2—Latency of self-grooming (s)	-	-	-	-	-	*	-
OF2—Total horizontal activity (n crossings)	*	*	-	-	-	-	-
OF2—Total horizontal activity in the center (n)	*	-	-	-	-	-	-
OF2— Total horizontal in the periphery (n)	-	*	-	-	-	-	-
OF2—Total vertical activity (n rearings)	-	*	-	-	-	*	-
OF2—Urination (%)	-	-	***	-	-	-	-
OR Sample trial—Time exploring object (s)	-	*	-	-	-	-	-
OR-Sample trial—Time to reach the criteria (s)	-	-	-	-	-	-	*
OR—Test trial—Object latency (s)	*	-	-	-	-	-	**
Marble test (MB)							
Intact (n)	-	*	-	-	***	-	-
Buried (n)	-	***	-	-	*	-	-
Spontaneous alternation in the T-Maze test							
Latency of first movement (s)	-	***	-	*	-	-	-
Latency to cross the intersection (s)	-	*	*	-	-	-	-
Total time to complete the test (s)	-	-	-	-	-	-	-
Total number of errors (n)	*	**	-	-	-	-	-
T-Maze test							
Latency of first movement (s)	*	-	-	-	-	-	-
Test trial—Latency to cross the intersection (s)	**	-	-	-	-	-	-
Test trial—Time to complete the test (s)	**	-	-	-	-	-	-
Morris water maze test (MWM)							
Escape latency—CUE	-	-	-	-	-	-	-
Escape latency—PT1	-	-	-	-	-	-	-
Escape latency—PT2	-	-	-	-	-	-	-
Escape latency—PT3	**	**	-	-	-	-	-
Escape latency—PT4	*	***	-	-	-	-	-
Distance—CUE	-	-	*	-	-	-	-
Distance—PT1	-	-	-	-	-	-	-
Distance—PT2	-	-	-	-	-	-	-
Distance—PT3	-	**	-	-	-	-	-
Distance—PT4	-	***	-	-	-	-	-
Swimming speed—CUE	***	***	-	**	-	-	*
Swimming speed—PT1	***	**	-	-	-	-	-
Swimming speed—PT2	**	**	-	-	-	-	-
Swimming speed—PT3	-	***	-	-	-	-	-
Swimming speed—PT4	*	**	-	-	-	-	*
Probe trial—Opposite quadrant distance (m)	-	***	-	-	*	-	-
Probe trial—Target quadrant distance (m)	*	-	-	-	-	-	*
Probe trial—Latency to platform (s)	*	-	-	-	-	-	-
Physical status							
Body weight (g)	-	***	***	-	-	*	-
Frailty index	***	***	*	-	-	-	-

**Table 2 ijms-24-13796-t002:** Age-dependent progression of Aβ intracellular signal and plaques and tau phosphorylation in male and female 3xTg-AD mice (n = 4/age-sex group). % of mice presenting positive cell markers (100% = 4/4 animals, 75% = 3/4 animals, 50% = 2/4 animals, 25% = 1/4 animals, 0% = 0/4 animals). PFC: the prefrontal cortex, AMG: amygdala, HCd: the dorsal hippocampus. HCv: ventral hippocampus and ECx: entorhinal cortex.

% Mice with Positive Signal										
Sex	Males					Females				
Brain Region	PFC	AMG	HCd	HCv	ECx	PFC	AMG	HCd	HCv	ECx
*6E10 Intracellular signal*										
2 months	100	100	100	100	100	75	75	75	75	75
4 months	100	100	100	100	100	100	100	100	100	100
6 months	100	100	100	100	100	100	100	100	100	100
12 months	100	100	100	100	100	100	100	100	100	100
16 months	100	100	75	100	75	100	100	75	100	75
*6E10 Aβ plaques*										
2 months	0	0	0	0	0	0	0	0	0	0
4 months	0	0	0	0	0	0	0	0	0	0
6 months	0	0	0	0	0	0	0	0	0	0
12 months	0	0	0	0	0	0	0	25	100	50
16 months	0	0	25	75	50	25	25	25	25	25
*AT8 Intracellular signal*										
2 months	25	100	50	100	75	0	100	0	100	100
4 months	50	75	50	75	50	50	75	50	75	75
6 months	25	50	25	50	50	50	100	75	100	100
12 months	50	75	75	75	100	75	100	100	100	100
16 months	100	100	100	100	100	50	100	75	50	50

## Data Availability

Available upon request. The original contributions presented in the study are included in the article; further inquiries can be directed to the corresponding author/s.

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
