# Peer review of "Survival Bias, Non-Lineal Behavioral and Cortico-Limbic Neuropathological Signatures in 3xTg-AD Mice for Alzheimer’s Disease from Premorbid to Advanced Stages and Compared to Normal Aging"

_ijms, 2023, doi:10.3390/ijms241813796_

Round 1

Reviewer 1 Report

Research in human aging needs to be improved by the scarcity of studies modeling its heterogeneity and complexity forged by pathophysiological conditions throughout the life cycle. Significant progress is emerging from the attention to two essential (interconnecting) factors: bilaterality of the human brain and prevalent chirality at the molecular levels. Deviation from the natural non-equilibrium molecular chirality due to abnormal time-dependent racemization contributes to protein aging.

               Any animal model of human physiology (in healthy and disease conditions) is reasonable only under focusing on the two factors mentioned above. Attempt to move forward without attention to fundamental features of animal physiology are misleading.

Author Response

ANSWERS TO REVIEWER 1

Research in human aging needs to be improved by the scarcity of studies modeling its heterogeneity and complexity forged by pathophysiological conditions throughout the life cycle. Significant progress is emerging from the attention to two essential (interconnecting) factors: bilaterality of the human brain and prevalent chirality at the molecular levels. Deviation from the natural non-equilibrium molecular chirality due to abnormal time-dependent racemization contributes to protein aging.

               Any animal model of human physiology (in healthy and disease conditions) is reasonable only under focusing on the two factors mentioned above. Attempt to move forward without attention to fundamental features of animal physiology are misleading.

RE: We’d like to thank the reviewer for his/her time to review our work and provide his/her opinion on research in human aging. However, the purpose of the present study is to provide a phenotypical and neuropathological characterization of an animal model, accepted by scientific community as listed in Alzforum. As for any animal model, scientific community is aware of the strong gaps and limitations of any animal model, but they are assumed and taken into account.

Reviewer 2 Report

The authors present a very interesting research article focusing on the implications of factors such as age, sex and phenotype towards development of the right animal model for studying neuropathological mechanisms involved with neuropsychiatric symptoms related to Alzheimer’s disease. Herein the authors conducted a very comprehensive study, and evaluated the roles of age, AD-genotype, and sex in the behavioral tests’ battery in 3xTg- AD male and female mice at ages from 2-16 months old, depicting different stages of the disease. The comparisons were done with control non-transgenic mice with normal aging. Considering the bottlenecks associated with the existing animal models to recapitulate all AD characteristics, this topic is indeed very relevant and would certainly help towards development of more therapeutic strategies in the future. The overall study approach and experimental strategy adopted for the whole study needs to be appreciated. Authors have incorporated an extensive array of testing with different groups of animals which has also helped collecting immense amount of data. The collected data have also been discussed in detail with appropriate justification using literature support.
There are some minor concerns noted below; which can be addressed by the authors in a straight forward manner following which the manuscript can certainly be considered for acceptance.  

1.     Introduction, page 2, line 43: ‘no cure exists’ – is probably not true considering the very recent approval of a drug for AD. I would suggest the authors to slightly reword this statement.

2.     Lines 59-61 in introduction, page 2:  “Moreover, periphery crosstalk interaction may be related to AD development and progression. Brain oxidative stress, vascular dysfunction, and neuroimmunoendocrine crosstalk dysregulation have been described”- periphery crosstalk interactions need to be explained briefly for clarity. The second statement also seems to be too vague and needs to be rewritten pointing towards the major findings of those papers.

3.     Section 4.4, page 21, immunofluorescence analysis methodology- this section needs to describe the number of samples subjected to the analysis in greater detail. Considering the semiquantitative nature of the ImageJ analysis, it is important to describe this. How many sections per tissue were analyzed? How many fields per section were considered?

4.     Describe the rationale behind considering these particular age groups for animals? Was this consideration of ‘asymptomatic, prodromal, onset, advanced and very old’ age groups based on literature?

5.     Section 4.2 on behavioral assessments – ‘n=14-21 mice per each genotype and age experimental group’- since the ‘n’ among different groups were not constant and was variable, how was this factor considered for data analysis via statistical methods?

6.     Page 7, line 185: “The age effect was observed with a progressively increased of buried marbles until 12 months”- grammatically incorrect. Please rectify.

7.     All figure legends must mention the accurate ‘n’ of samples along with the statistical indications.

8.     Section 2.3, lines 287-290: “On the other hand, extracellular Aβ plaques were absent in the first stages of the disease. They were first detected in females of 12-month-old mice and in 16-month-old males, being firstly ventral hippocampus and secondarily entorhinal cortex”- was this detected in all females and males of these age groups? Or a certain subpopulation among this number?

9.     Table 2 is slightly confusing and needs to be better shown or explained with reference to number of animals for ease in understanding.

10.   The limitations of the current study need to be mentioned in the discussion section.

11.   One suggestion is to consider modifying the title of the manuscript to a more concise statement.

Only minor correction needed as indicated in my review comments. 

Author Response

ANSWERS TO REVIEWER 2

We’d like to thank you the reviewer for his/her time to review our work and provide a number of suggestions, comments and good advice to improve the quality of it. Here we provide a point-by point answer to all questions. Changes in the Ms. have been indicated with yellow.

The authors present a very interesting research article focusing on the implications of factors such as age, sex and phenotype towards development of the right animal model for studying neuropathological mechanisms involved with neuropsychiatric symptoms related to Alzheimer’s disease. Herein the authors conducted a very comprehensive study, and evaluated the roles of age, AD-genotype, and sex in the behavioral tests’ battery in 3xTg- AD male and female mice at ages from 2-16 months old, depicting different stages of the disease. The comparisons were done with control non-transgenic mice with normal aging. Considering the bottlenecks associated with the existing animal models to recapitulate all AD characteristics, this topic is indeed very relevant and would certainly help towards development of more therapeutic strategies in the future. The overall study approach and experimental strategy adopted for the whole study needs to be appreciated. Authors have incorporated an extensive array of testing with different groups of animals which has also helped collecting immense amount of data. The collected data have also been discussed in detail with appropriate justification using literature support.
There are some minor concerns noted below; which can be addressed by the authors in a straight forward manner following which the manuscript can certainly be considered for acceptance.  

  1. Introduction, page 2, line 43: ‘no cure exists’ – is probably not true considering the very recent approval of a drug for AD. I would suggest the authors to slightly reword this statement.

RE: Thanks for the comment. We have modified the sentence as “there is a high failure rate in developing new disease-modifying treatments”

  1. Lines 59-61 in introduction, page 2:“Moreover, periphery crosstalk interaction may be related to AD development and progression. Brain oxidative stress, vascular dysfunction, and neuroimmunoendocrine crosstalk dysregulation have been described”- periphery crosstalk interactions need to be explained briefly for clarity. The second statement also seems to be too vague and needs to be rewritten pointing towards the major findings of those papers.

RE: Thanks for the comment. We have modified the sentence as “Moreover, although the connections are not yet fully understood, there is growing experimental, clinical, and epidemiological evidence that brain and periphery crosstalk interaction may be related to AD development and progression. Underlying systemic disease processes are reflected in AD patients: Systemic immunity disorders, cardiovascular disease, hepatic and renal dysfunction, metabolic disorders, blood abnormalities, respiratory and sleep disorders, microbiota disorders, and systemic inflammation”

  1. Section 4.4, page 21, immunofluorescence analysis methodology- this section needs to describe the number of samples subjected to the analysis in greater detail. Considering the semiquantitative nature of the ImageJ analysis, it is important to describe this. How many sections per tissue were analyzed? How many fields per section were considered?

RE: Thanks for the comment. Three different (added in line 683) tissue sections were analyzed (1 section from bregma +1.5 for PFC, 1 section from bregma -1.5 for AMG and HCd and 1 section from bregma -3 for HCv and ECx). Just one field per section was analysed for that reason we explained that is a semiquantitative analysis. We added in the limitations that a more extensive analysis with more fields per sections will be needed.

  1. Describe the rationale behind considering these particular age groups for animals? Was this consideration of ‘asymptomatic, prodromal, onset, advanced and very old’ age groups based on literature?

As described in Alzforum for the phenotypical characterization of this animal model, the consensus given (as supported literature) is:

  1. Section 4.2 on behavioral assessments – ‘n=14-21 mice per each genotype and age experimental group’- since the ‘n’ among different groups were not constant and was variable, how was this factor considered for data analysis via statistical methods?

RE: Thanks for the comment. We have added the sample size per each genoype/age. According to guidelines (Gerlai et al., 1999) for mutants a sample size of 10 is recommended. We have from 14 to 21, and sphericity (in ANOVA) or variance (t-tests) is taken under consideration when providing the statistical result.

  1. Page 7, line 185: “The age effect was observed with a progressively increased of buried marbles until 12 months”- grammatically incorrect. Please rectify.

RE: Thanks for the observation. We have modified the sentence as “An age-dependent progressive increase of buried marbles was observed until 12 months of age”

  1. All figure legends must mention the accurate ‘n’ of samples along with the statistical indications.

RE: Thanks for the comment. We have modified all the legends as requested: 2m-NTg (n = 16, 8 males and 8 females), 2m-3xTg-AD (n = 16, 8 males and 8 females); 4m-NTg (n = 16, 8 males and 8 females), 4m-3xTg-AD (n = 15, 7 males and 8 females); 6m-NTg (n = 21, 11 males and 10 females), 6m-3xTg-AD (n = 21, 11 males and 10 females); 12m-NTg (n = 17, 9 males and 8 females), 12m-3xTg-AD (n = 18, 9 males and 9 females); 16m-NTg (n = 14, 8 males and 6 females), 16m-3xTg-AD (n = 18, 9 males and 9 females).

  1. Section 2.3, lines 287-290: “On the other hand, extracellular Aβ plaques were absent in the first stages of the disease. They were first detected in females of 12-month-old mice and in 16-month-old males, being firstly ventral hippocampus and secondarily entorhinal cortex”- was this detected in all females and males of these age groups? Or a certain subpopulation among this number?

RE: Thanks for the observation. The new version explains the animals that were detected. “They were first detected in females of 12-month-old mice and in 16-month-old males, being firstly ventral hippocampus (100% of 12-month female and 75% of 16-month male) and secondarily entorhinal cortex (50% of 12-month female and 50% of 16-month male)”

  1. Table 2 is slightly confusing and needs to be better shown or explained with reference to number of animals for ease in understanding.

RE: Thanks for the good advice. Explained in Table 2. % of mice presenting positive cell markers (100%= 4/4 animals, 75%= 3/4 animals, 50%= 2/4 animals, 25%= 1/4 animals, 0%= 0/4 animals).

  1. The limitations of the current study need to be mentioned in the discussion section.

RE: Thanks for the good advice. A specific section for limitations has been included in the discussion, and reads as follows: “ One of the limitations of the present study is the difficulties in obtaining a correct sample size of aged animals due to the higher heterogeneity in physiological and behavioral heterogeneity presented in pathological and non-pathological aging processes. Interlaboratory reliability would be interesting to study as the generation of sublines with different onset and progression of symptoms have been described, even between littermates. On the other hand, a more accurate histological study should be interesting to perform with an increased number of sections per tissue analyzed to quantify the neuropathology damage better, also with other more quantitative techniques.”

  1. One suggestion is to consider modifying the title of the manuscript to a more concise statement.

RE: Thanks for the good advice. We have shortened the title of the manuscript as requested.
